# *Mimulus* sRNAs Are Wound Responsive and Associated with Transgenerationally Plastic Genes but Rarely Both

**DOI:** 10.3390/ijms21207552

**Published:** 2020-10-13

**Authors:** Jack Colicchio, John Kelly, Lena Hileman

**Affiliations:** 1Department of Plant and Microbial Biology, University of California, Berkeley, CA 94720, USA; 2Ecology and Evolutionary Biology, University of Kansas, Lawrence, KS 66045, USA; jkk@ku.edu (J.K.); lhileman@ku.edu (L.H.)

**Keywords:** small RNAs, epigenetics, transgenerational plasticity, wound-response, tRFs, *Mimulus guttatus*

## Abstract

Organisms alter development in response to environmental cues. Recent studies demonstrate that they can transmit this plasticity to progeny. While the phenotypic and transcriptomic evidence for this “transgenerational plasticity” has accumulated, genetic and developmental mechanisms remain unclear. Plant defenses, gene expression and DNA methylation are modified as an outcome of parental wounding in *Mimulus guttatus*. Here, we sequenced *M. guttatus* small RNAs (sRNA) to test their possible role in mediating transgenerational plasticity. We sequenced sRNA populations of leaf-wounded and control plants at 1 h and 72 h after damage and from progeny of wounded and control parents. This allowed us to test three components of an *a priori* model of sRNA mediated transgenerational plasticity—(1) A subset of sRNAs will be differentially expressed in response to wounding, (2) these will be associated with previously identified differentially expressed genes and differentially methylated regions and (3) changes in sRNA abundance in wounded plants will be predictive of sRNA abundance, DNA methylation, and/or gene expression shifts in the following generation. Supporting (1) and (2), we found significantly different sRNA abundances in wounded leaves; the majority were associated with tRNA fragments (tRFs) rather than small-interfering RNAs (siRNA). However, siRNAs responding to leaf wounding point to Jasmonic Acid mediated responses in this system. We found that different sRNA classes were associated with regions of the genome previously found to be differentially expressed or methylated in progeny of wounded plants. Evidence for (3) was mixed. We found that non-dicer sRNAs with increased abundance in response to wounding tended to be nearby genes with decreased expression in the next generation. Counter to expectations, we did not find that siRNA responses to wounding were associated with gene expression or methylation changes in the next generation and within plant and transgenerational sRNA plasticity were negatively correlated.

## 1. Introduction

The capacity of cells carrying identical DNA sequences to function in remarkably divergent ways is vital for the evolution of complex life. To achieve complex forms, varied developmental and physiological programs are initiated across organs, tissues and cells in response to immediate developmental or environmental cues. These programs can leave a lasting epigenetic mark on the genome in cells that have received developmental or environmental signals. These lasting epigenetic modifications include DNA methylation, histone methylation and histone acetylation and can alter the RNA expression, chromatin accessibility and splicing that occurs proximal to these markings [1,2,3]. Additionally, these markings can be transmitted between mitotic and sometimes meiotic cell divisions [4,5] allowing developmental and environmental cues to leave a lasting impact on a cell lineage, altering the development of cells in later life history stages or even in the next generation.

Meiotic transmission of epigenetic markings resulting from environmental cues provides a mechanism through which plastic responses to the environment can be transmitted from one generation to the next (transgenerational plasticity [6]). While plant germline cells do undergo a number of rounds of epigenetic reprogramming prior to the formation of a new seed [7], it is clear that environmental conditions experienced by parents can alter phenotypes [8,9,10,11], epigenetic profiles [4,10,12,13] and gene expression [14,15,16] in the next generation. Importantly, autocorrelation between offspring and parent environment can favor mechanisms that establish such patterns of transgenerational plasticity [17,18], perhaps leading to natural selection that either favors or disfavors this transmission depending on local environmental patterns [11].

A well-developed model for meiotic transmission of epigenetic markings in plants is the loading of small RNAs (sRNA) into the germline during or after somatic epigenetic reprogramming [5]. These sRNAs may affect germline histone methylation/acetylation and/or DNA methylation status [19] which, in turn, when transmitted to the next generations, affects patterns of gene expression and developmental and/or physiological output [5,20,21]. Germline methylation is implicated in transgenerational plasticity because experimental de-methylation alters transgenerational responses [9,10,22]. However, for developmentally plastic responses to environmental cues in one generation to lead to altered development in the next generation, mobile epigenetic modifiers such as sRNAs that are capable of influencing the epigenetic state of the germline and are stress responsive [23,24] are necessary. In other taxa this has been demonstrated; in *C. elegans* sRNAs are fundamental to the transmission of epigenetic information between generations [25]. Small RNAs are known to modify patterns of germline DNA methylation in plant models [23], yet it is unknown the extent to which they facilitate transgenerational environmental responses.

Plants produce a diverse array of sRNAs including micro RNA (miRNA), transfer RNA (tRNA), ribosomal RNA (rRNAs), small non-coding RNA (snRNAs) and dicer-derived small interfering RNA (siRNA) [21,26]. tRNAs and rRNAs have vital functions across eukaryotes and miRNAs are critical in developmental and environmental responses. However, siRNAs are directly involved in altering histone modifications and DNA methylation in plants [27,28]. Plant siRNAs are processed from double stranded precursors by dicer proteins, leading to a specific size of siRNAs between 20 and 24 nt being produced [29]. These siRNAs can function in epigenetic repression through both transcriptional and post-transcriptional silencing depending on their size class.

Plant siRNAs, especially 24 nt siRNAs, are particularly intriguing as candidates for the transmission of epigenetic signals between generations. They are stress-responsive [30,31], able to mobilize across plant tissues [32,33,34], alter DNA methylation [35,36] and are able to enter into the germline [36,37]. In *Brachypodium*, 24 nt siRNAs derived from 3′-UTRs are up-regulated in response to abiotic stresses and target intronic regions of many messenger RNAs in *trans*, potentially impacting splicing and expression of these genes [38]. These 24 nt siRNAs are produced by RNA Pol IV and direct DNA methylation in a sequence specific fashion in *Arabidopsis* [28,39], rice [40] and other species [41,42]. Additionally, siRNAs produced in the pollen vegetative cell are loaded into the sperm cell leading to the silencing of transposable elements [43,44], potentially altering DNA methylation in the zygotes after fertilization [45]. The combination of this trio of features grants 24 nt siRNAs the suite of properties necessary to mediate transgenerational plasticity. Indeed, in response to cold, RNA-dependent DNA Methylation (RdDM) in the paternal sperm cell and maternal central cell at the *ALLANTOINASE* (*ALN*) locus increases DNA methylation, reduces gene expression and promotes a highly dormant seed state [46].

tRNA derived RNA fragments (tRFs) are another class of stress responsive sRNAs that can function in transcriptional, post-transcriptional and translational regulation [47]. tRFs are diverse and ubiquitous across all domains of life [48]. They range in size from 15-35 base pairs, encompassing both dicer-dependent and independent molecules which can be derived from the 5′ or 3′ end of the tRNA. In Arabidopsis, tRF expression has been shown to be highly plastic to UV, Cold, Drought and Salt stress, as well as across plant tissues [49]. Saliently, these tRF sRNAs have recently been shown to contribute to transgenerational inheritance in mice [50,51,52]. They are loaded into sperm cells and subsequently influence diet-induced gene expression changes in progeny. In plants, their role in transgenerational plasticity is less clear but recent work in *Arabidopsis* has demonstrated that parental heat stress leads to significant shifts in tRF profiles. Notably, decreased expression of certain isoacceptors and increased expression of others, such as tRNA-Asp in the progeny of heat stressed plants [53]. These results suggest that even sRNAs not typically associated with RNA interference pathways may play a vital role in mediating transgenerational plasticity.

*Mimulus guttatus* is a well-known system for studies in ecological and evolutionary genetics and in recent years has also become a plant model for transgenerational plasticity in response to wounding. Offspring of leaf-wounded parents exhibit phenotypic [9,54,55], gene expression [14,56] and DNA methylation [13] differences compared to control plants. There is substantial genetic variation within *M. guttatus* for the transgenerational induction of trichomes [54,55] but the mechanisms that mediate this response remain unclear. While the molecular resources available within the *M. guttatus* system are still limited compared to systems such as *Arabidopsis*, they have grown dramatically in recent years [57]. Striking morphological and genetic diversity, well-studied phenotypic plasticity and ecologically relevant transgenerational plasticity make *M. guttatus* an ideal model for studying epigenetic plasticity. Additionally, since transgenerational plasticity is expected to be highly variable across taxa [18], it is important that numerous model systems are developed in order to fully understand the diversity of mechanisms through which organisms mediate transgenerational plasticity. To date, neither histone modifications, nor sRNAs have been investigated in the *M. guttatus* system, both of which are candidates in mediating within and between generation plasticity to environmental cues.

To investigate a potential role for sRNAs in transgenerational plasticity, we surveyed sRNA abundances in response to leaf wounding within and between generations. We hypothesized that sRNAs produced in response to wounding elicit the epigenetic, transcriptomic and ultimately phenotypic transgenerational responses observed in *M. guttatus*. From the literature described above, our *a priori* model was that sRNAs produced in response to wounding are responsible for changing the epigenetic regulatory landscape leading to a combination of altered sRNA, methylation and gene expression levels in the following generation. To test this model, we determined (1) whether siRNA (or other sRNA) populations differ between leaf-wounded and control plants; (2) whether proximity of sRNA loci increases the probability that protein coding regions will be differentially methylated or differentially expressed in progeny of leaf wounded plants; and (3) whether sRNA abundance, gene expression or DNA methylation differences in offspring of wounded compared to control plants correlate with sRNA profiles in the parent generation after leaf wounding.

## 2. Results

### 2.1. Classification of the Mimulus Leaf sRNAome

We sequenced 34 leaf sRNA libraries and identified 29,884 sRNA loci (Figure 1a). In total, we sequenced ca. 230 million sRNA fragments, of which ca. 179 million mapped to our RIL94 genome, with an average of 5.2 million reads mapped per sample (3.1–11.4 million reads). We partitioned the loci according to whether they produced sRNAs that likely function in the RNAi pathway (classified as dicer-derived sRNA, siRNAs), whether they likely function as micro RNAs (miRNAs) or neither (non-dicer sRNAs, nd-sRNAs) (Figure 1b, Appendix A). We further divided siRNA loci according to the size class of sRNAs produced (Table 1).

The expression of sRNAs from these loci varied significantly by class (df = 6, F = 457, *p* < 0.00001). miRNA loci produced the largest number of reads, averaging (log10(rpm) = 1.4) and 24 nt siRNA loci the fewest, averaging (log10(rpm) = 0.36, Table 1, Appendix A). Locus length also varied significantly by class (df = 6, F = 553, *p* < 0.0001). nd-sRNA loci were on average the longest (774 bp), with miRNA (227 bp) and 20 nt siRNA (200 bp) the shortest (Table 1). The location of sRNA loci can shed light on their role in transposon and/or gene regulation and as expected 90.4% of the 24 nt siRNA loci overlapped with annotated transposable elements, compared to 51.6% of 21 nt siRNA and 30.8% of nd-sRNA loci (Table 1). 62.7% of nd-sRNA loci overlapped with annotated protein coding genes, while 21 nt siRNA loci overlapped 45.3% of the time; other sRNA loci overlapped with protein coding genes less than 13% of the time. 24 nt siRNA loci rarely overlapped with protein coding regions but 67.5% were found within 5 kb of protein coding regions, potentially within gene regulatory sequences (Table 1, Appendix A).

### 2.2. Wound Responsive sRNAs

To test component (1) of our a priori model, whether siRNA (or other sRNA) populations differ between leaf-wounded and control plants, we identified sRNAs produced in different abundance between wounded and damaged leaf tissues (Figure 1c). At the 1 h and the 72 h time-points in the second leaf pair, we found that 3 miRNA, 0 siRNA and 53 nd-sRNA loci, 10 miRNA, 5 siRNA and 377 nd-sRNA loci showed significantly different sRNA abundances in leaf damaged compared to control plants, respectively (FDR < 0.05, Figure 2). Only one miRNA locus and no other sRNA loci showed significantly different sRNA abundances in the unwounded third leaf pair in leaf damaged compared to control plants. At the second leaf pair of leaf wounded offspring compared to offspring of control plants, we identified 2 miRNA, 1 siRNA and 1 nd-sRNA locus with significantly different sRNA abundances (FDR < 0.05).

The most striking pattern of differential sRNA abundances in response to leaf wounding was for nd-sRNA loci in the wounded leaves themselves. In wounded leaves, approximately 7 times more loci exhibited differential sRNA production 72 h after wounding compared to 1 h after wounding. We found extensive overlap between the 1 h and 72 h sets of differentially expressed loci; 50/53 nd-sRNA loci that showed a pattern of differential sRNA production at 1 h retained that pattern 72 h after leaf wounding. Of the 377 nd-sRNA loci differentially expressed at the 72 h time-point, 328 loci showed a pattern of increased sRNA abundance and 49 of decreased sRNA abundance in response to leaf wounding. The lack of differential sRNA abundances in the third leaf pairs, which did not experience wounding, suggests that while sRNA responses in wounded leaves are extensive, the mobility of these sRNAs into adjacent tissues is limited. It is however possible that sRNAs produced at the site of leaf wounding are transiently present in adjacent tissues or in the next generation and even at low levels are capable of altering DNA methylation or histone markings even when they are no longer detectable.

#### 2.2.1. Micro RNA (miRNA) Loci

Of the three miRNA loci that showed a pattern of differential sRNA production 1 h after wounding, two showed a pattern of increased sRNA production (miRNA-169b and miRNA164a) and the third showed a pattern of decreased sRNA production. At the 72 h time-point in wounded 2nd pair leaves, we found that 10 miRNA loci showed a pattern of differential sRNA production. Eight of the 10 increased sRNA levels while 2 of the 10 decreased sRNA levels in response to leaf wounding. Both miRNA loci with decreased sRNA production in response to leaf wounding belonged to the same class (miRNA166). The 8 miRNA loci with increased sRNA production were a miRNA167, miRNA133, miRNA399, miRNA157, 2 miRNA169s and 2 miRNA156s. In the adjacent, unwounded 3rd leaf pair, only a miRNA160 locus shows differential sRNA production in response to 2nd leaf pair wounding. And in the next generation 1 miRNA169 showed a pattern of decreased sRNA production in response to leaf wounding in the parental generation.

We identified the putative gene targets of differentially regulated miRNA loci. The most strongly differentially expressed locus (miRNA_166_mg1, *p* = 0.000047) exhibited elevated sRNA production in the 2nd leaf pair 72 h after leaf wounding compared to control and its putative target genes were enriched for leucine-rich repeat family proteins (*p* = 9.0 × 10^−7^) and pumillo family proteins (*p* = 5.5 × 10^−7^). Interestingly, the fourth most strongly differentially expressed miRNA locus in the 2nd leaf pair at 72 h after leaf wounding was also the only miRNA locus significantly differentially expressed in the progeny of wounded plants. At the 2nd leaf pair in the parent generation, this miRNA locus showed a pattern of increased sRNA production in leaf damaged compared to control tissues while in the progeny generation, this miRNA locus showed a pattern of decreased sRNA production in response to parental leaf damage. Putative targets of sRNAs from this miRNA locus are highly enriched for FAD (Flavin adenine dinucleotide) binding proteins (*p* = 3.5 × 10^−5^) which are involved in plant stress responses and secondary metabolism [58]. That we see the direction of sRNA production from this miRNA locus reverse between parental to offspring generations is unexpected, yet this pattern arises again for other classes of sRNA loci (see tRFs below).

#### 2.2.2. Dicer-Derived sRNA (siRNA) Loci

siRNA loci are the most abundant in our dataset, yet we only found 6 siRNA loci with a pattern of differential abundances in response to leaf wounding. Five in the comparison between wounded and control 2nd leaf pair tissue at the 72 h time-point (1 22 nt, 2 23 nt, 2 24 nt siRNAs) and one in the comparison between offspring of leaf wounded compared to control parents (a 24 nt siRNA). These results suggest that differential sRNA production from siRNA loci in response to wounding is limited, at least in the tissue types assayed here.

In the parental generation two of the differentially expressed siRNA loci (cluster: 10,399, *p* = 5.9 × 10^−36^, LFC = 2.7: cluster: 10,398, *p* = 1.8 × 10^−6^, LFC = 1.9) overlapped with the same gene, Migut.F00689 homologous to Arabidopsis Jasmonic Acid Oxidase (JOX1, AT3G11180.1, 81% amino acid similarity). Repression of this gene likely gives rise to elevated Jasmonic Acid production, making it a prime candidate for induction of wound responses both within and between generations. In terms of differentially expressed siRNAs produced in response to wounding that may function in regulating epigenetic marks, a 23 nt siRNA locus overlapping with Migut.L02022 (Histone H4) (cluster:24458, *p* = 5.7 × 10^−8^, LFC = 2.6) had significantly increased sRNA production in response to leaf wounding at the 72 h 2nd leaf pair time-point. Additionally, although they did not pass genome wide significance, an siRNA locus overlapping with Migut.B01043 (H3K9 demethylase, cluster:2545, *p* = 0.0002, LFC = 1.2) nearby (1.9 kb away) another H3K9 demethylase (Migut.H00959, cluster 14258, *p* = 0.003, LFC = 1.3) showed a pattern of increased siRNA production in 2nd pair leaves 72 h after wounding. Recent work has demonstrated a key role of H3K9 methylation in epigenetic responses to stress and suggested a potential key role for H3K9 modifications in transgenerational plasticity [59,60]. The only siRNA locus differentially expressed in the progeny of wounded plants compared to control plants was a 24 nt siRNA locus with elevated sRNA production in the progeny of wounded plants; this siRNA locus overlaps with a homolog of Calmodulin, Migut.N02025 (cluster 28767, *p* = 1.8 × 10^−7^, LFC = 2.81). Calmodulin has a known regulatory role in plant abiotic responses through modifying calcium signaling in the chloroplast [61]. This provide intriguing evidence for a role of calcium and/or protein transport between the cytoplasm and chloroplast (retrograde or anterograde signaling) in the Mimulus transgenerational responses to wounding.

#### 2.2.3. Non-Dicer Derived sRNA (nd-sRNA) Loci

Nd-sRNA loci most often exhibited differential sRNA abundances in response to leaf wounding compared to other sRNA categories. Most of these differentially expressed nd-sRNA loci did not overlap with protein coding genes (315/377 nd-sRNA loci differentially expressed in 2nd pair leaves at the 72 h time-point did not overlap). Of the 62 wound-responsive nd-sRNA loci that overlapped with protein coding regions, there was significant enrichment for NADH:quinone oxidoreductase, which can bind Calcium ions and functions in cell respiration [62]. Only one nd-sRNA locus showed a pattern of differential sRNA production in the offspring of wounded compared to control parents. This nd-sRNA locus was down-regulated (*p* = 0.00000155, LFC = −3.2) in the progeny of wounded plants and is located in the coding region of Migut.K01186, a homolog of Arbidopsis Ycf2, which is involved in chloroplast protein uptake [63]. Additionally, we used an LFC outlier enrichment approach similar [64] and found the following. Nd-sRNA loci with decreased sRNA abundances in response to parent leaf wounding were enriched in their overlap with Heat Shock protein coding sequences (*p* = 2.7 × 10^−6^). Nd-sRNA loci with increased sRNA abundances in response to parent leaf wounding were enriched in their overlap with chlorophyll A-B binding protein coding sequences (*p* = 1.4 × 10^−14^).

The majority of wound-responsive nd-sRNA loci did not overlap with protein coding regions. Instead, 34 of the top 40 differentially expressed nd-sRNA loci overlapped tRNA loci (detected using the tRNADB-CE database). Small RNAs mapping to tRNA loci ranged in size from 19–35 nt and represent tRNA-derived small RNA fragments (termed tRFs). Twenty three of the 34 loci produced tRFs that mapped to the 5′ end of tRNAs, including the D-loop; 11 of the 34 loci produced sRNAs that mapped to the 3′ end of the anicodon, including the variable loop, T loop and acceptor stem. None of the tRFs mapped to tRNA anticodon sequences. Many tRFs mapped to different copies of the same tRNA anticodon type; in total, we found 12 different tRNAs to which tRFs mapped. The tRNA loci with the strongest pattern of differential tRF production in response to leaf wounding were Leucine tRNA (CAA) (*p* = 7.6 × 10^−18^) followed by Glycine tRNA (CCC)(1.4 × 10^−13^) (Figure 3a). The Alanine tRNA (GGC) locus was unique among the tRNAs producing differential tRF abundances in response to leaf wounding in that tRFs mapping to this locus were significantly more abundant in the 2nd leaf pair 1 h (*p* = 1.7 × 10^−10^), as well as 72 h (*p* = 9.8 × 10^−13^) after leaf wounding (Figure 3a).

#### 2.2.4. Response Correlations at nd-sRNA Loci

We tested for correlations in sRNA abundance shifts at various time points after wounding and into the next generation. Log-fold changes (LFC) of nd-sRNA abundances in response to leaf wounding are positively correlated between the wounded leaf 1 h and 72 h after wounding (r = 0.272). On the other hand, LFCs of nd-sRNAs at the 2nd wounded leaf pair 72 h after wounding were negatively correlated with changes in nd-sRNA abundances at the 3rd un-wounded leaf pair 72 h after 2nd leaf pair wounding (r = −0.249). Additionally, LFCs of nd-sRNAs at the 2nd leaf pair 72 h after wounding were also negatively correlated with 2nd leaf pair nd-sRNA LFCs in the offspring generation (r = −0.202, Appendix A). siRNA responses across developmental time-points within the wounded generation were positively correlated, albeit less significantly than nd-sRNAs (0.034 < r < 0.109, Appendix A). siRNA responses were negatively correlated between parent and offspring generations (−0.096 < r < −0.023, Appendix A). Negative correlation of sRNA abundances in response to leaf wounding between 2nd and 3rd pair leaves in the parent generation (nd-sRNA loci) and between 2nd pair leaves in the parent and offspring generations (nd-sRNA and siRNA loci) suggests that transgenerational plasticity is not likely the result of passive diffusion or transmission of wound induced sRNAs from parent to offspring.

tRNA-derived tRF sRNAs show a striking pattern of correlated change in response to leaf wounding. tRNA loci with elevated tRF sRNA abundance in the 2nd leaf pair 72 h after wounding show a clear pattern of reduced tRF abundance in the progeny of wounded plants (mean LFC: −0.32, sd = 0.41, t = −13.96, *p* < 0.0001, *n* = 315, Figure 3b,c). However, tRNA loci with elevated tRF sRNA abundance in the 3rd leaf pair 72 h after wounding show a clear pattern of elevated tRF abundance in the progeny of wounded plants (r = 0.23, t = 4.33, *p* = 0.00002, *n* = 315, Figure 3c). As expected, when modeled together we found a negative effect of wounded second leaf pair (*p* < 0.0001) and a positive effect of third leaf pair LFC (*p* = 0.00006) on offspring LFC for this set of 315 sRNAs.

### 2.3. Relationship between sRNA Loci and Transgenerationally Plastic DNA Methylation and Gene Expression

Previous experimental work in this same M. guttatus RIL94 genotype identified 3731 differentially methylated genomic regions (DMRs) [13] and 919 differentially expressed genes (DEGs) [14] in the progeny of leaf wounded plants. To test component (2) of our working model, whether proximity of sRNA loci increases the probability that protein coding regions will be differentially methylated or differentially expressed in progeny of leaf wounded plants, we tested whether genomic regions that overlapped with sRNA loci were more likely to have previously been identified as transgenerationally labile. To test component (3) of our model, whether sRNA abundance, gene expression or DNA methylation differences in offspring of wounded compared to control plants correlate with sRNA profiles in the parent generation after leaf wounding, we tested whether LFCs in sRNA abundance were predictive of the direction of DNA methylation change or gene expression change in the progeny of leaf wounded plants.

#### 2.3.1. sRNA Loci and Differential Methylation

Global DNA methylation levels at protein coding regions (independent of leaf wounding treatment) were associated with variation in the presence of sRNA loci. Protein coding regions overlapping with siRNA loci tended to have higher levels of CG, CHG and CHH DNA methylation than background averages, while protein coding regions overlapping with nd-sRNA loci tended to have overall lower levels of DNA methylation (Appendix A). 24 nt siRNA loci overlapped with DMRs (in response to parent leaf wounding) more frequently than 20-23 nt siRNA loci or nd-sRNA loci (*p* < 0.0001, Chi Sq. = 67.58), even after accounting for variation in sRNA locus size (*p* < 0.0001, Chi Sq. = 152) and variation in position of sRNA loci with respect to protein coding regions (overlapping with, <5 kb from or >5 kb from a gene; *p* = 0.008, Chi Sq. = 9.8). For instance, 500 bp long siRNA loci that are within 5 kb of a protein coding region (Distance class: N, near) overlap with a DMR 5.6% of the time, compared to only 1.4% for nd-sRNA loci (Figure 4a). Additionally, sRNA class had a significant effect on the direction of methylation change in the offspring of wounded plants. DMRs that do not overlap with sRNA loci were up-methylated 48.8% of the time in response to parent wounding. DMRs that overlap with nd-sRNA loci were up-methylated 41.8% of the time in response to parent wounding and DMRs that overlap with siRNA loci were up-methylated 65.9% of the time in response to parent wounding (*p* < 0.001, Chi Sq. = 75.21, Figure 4b). This result demonstrates that siRNA loci tend to be up-methylated in the progeny of wounded plants, suggesting that RNA-dependent DNA methylation (RdDM) may play a role in the transmission of hypermethylated states during transgenerational plasticity. As expected from the high propensity of 24 nt siRNA loci that overlap with TEs as a whole (Table 1), the majority of siRNA-associated DMRs overlapped with TEs (79.7%) rather than protein coding genes (11.4%).

We detected significant overlap between sRNA loci and DMRs and further tested whether changes in sRNA abundance in response to wounding was associated with our previously observed direction of methylation change in the offspring of wounded parents. From our a priori model, we hypothesized a positive correlation between changes in RdDM associated 24 nt siRNA abundances in response to wounding and changes in DNA methylation states in the progeny generation. In other words, mobile siRNA signals produced in wounded tissues may be capable of programming methylation patterns in the next generation. After filtering to only include the 13,621 sRNA loci within 20 kb of a previously identified DMRs, we did not find evidence for association between progeny generation DNA methylation change and LFC at siRNA loci in either the 2nd or 3rd leaf pair of the parental generation (*p* > 0.1). However, we did find a significant association between progeny generation DNA methylation changes and LFC at nd-sRNA loci in the 3rd leaf pair of the parental generation (*p* = 0.009, Figure 4c). Fourteen of the 15 nd-sRNAs with the largest LFC increase in abundance at the 3rd leaf pair of the parent generation were within 20 kb of progeny generation up-methylated DMRs. These results are driven by nd-sRNAs that overlap protein coding regions of the genome, not the tRNA derived tRFs previously discussed. This result suggests that while 24 nt siRNAs are associated with DMRs, changes in their expression in wounded leaf tissue is not predictive of methylation changes in the following generation. However, there is a relationship between increased gene-associated nd-sRNA production and transgenerationally heritable increases in DNA methylation.

#### 2.3.2. sRNA Loci and Differential Gene Expression

We tested whether sRNA loci associated with previously identified differentially expressed genes (DEGs) in response to parental wounding. Overall we found that genes overlapping with nd-sRNA loci had significantly higher expression in response to parental wounding than genes not overlapping with sRNA loci and that genes overlapping with siRNA loci had significantly lower expression in response to parental wounding (Appendix A). This is in line with our finding that genomic regions overlapping with nd-sRNA loci tended to have lower methylation in response to parental wounding, while those regions overlapping with siRNA loci tended to have higher methylation in response to parental wounding. DNA methylation often signals gene repression [65], although the relationship between methylation and expression is clearly complex [66,67]. Specific gene ontology and protein domain categories were highly enriched in the different protein coding regions with overlapping sRNA loci. Genes overlapping with 21 nt siRNA loci were enriched for Adenosine diphosphate (ADP) binding (*p* = 2.50 × 10^−75^) and the interpro NB-ARC domain proteins (*p* = 7.89 × 10^−19^). Genes overlapping with 24 nt siRNA loci were enriched for sulfur compound transport (4.10 × 10^−07^) and cellulose metabolism (8.20 × 10^−07^) functions. Genes overlapping with nd-sRNA loci were enriched for chlorophyll binding (*p* = 4.1 × 10^−37^) and translation (*p* = 2.60 × 10^−61^). These results confirm that sRNA production is associated with distinct classes of protein coding genes.

Our initial model was that siRNAs produced in response to leaf wounding may modify gene expression transgenerationally through differential methylation, yet we find little evidence supporting this hypothesis. sRNAs produced in response to leaf wounding may modify expression patterns by other means such as histone modifications or direct sRNA transmission. Therefore, we tested whether overlap of protein coding regions with sRNA loci impacted the probability of differential expression in response to parental wounding, and, if so, whether sRNA loci were enriched in up- vs. down-regulated genes. After filtering out genes with very low expression, we found that the probability that a gene would be differentially expressed in response to parental wounding increased from 8.4% to 13.2% when overlapping with an sRNA locus.

Next, we included log(expression) and sRNA locus type (siRNA, nd-sRNA or neither) terms, with direction of change as the response variable. Both of these terms were significant (log(mean expression): **χ^2^** likelihood ratio = 722, *p* < 0.0001, sRNA overlap: X2LR: 42, *p* < 0.0001, Figure 5a). The significant log(mean expression term) revealed that genes with globally higher expression tended to be down-regulated in response to parental wounding and genes with globally lower expression tended to be up-regulated, which we have previously reported [14]. The significant sRNA locus type term demonstrated that genes overlapping with siRNA and nd-sRNA loci are more likely to be differentially expressed in response to parental wounding and revealed that different classes of sRNA loci were associated with different directions of transgenerational plasticity. Of genes with globally high levels of expression and overlapping directly with siRNA loci, 14.9% were down-regulated in the progeny of wounded plants. This is compared to only 9.1% and 6% of high expression level genes overlapping with nd-sRNA and no sRNA loci, respectively (Figure 5a) and 12.5% of genes with moderate global expression levels and overlapping with nd-sRNA loci were up-regulated in the progeny of wounded plants. This is compared to only 6.8% and 3.9% of moderately expressed genes overlapping with siRNA and no sRNA loci, respectively (Figure 5a). We found significant enrichment of heat-shock proteins in the set of genes that were up-regulated in response to parental wounding and overlapping with sRNA loci (*p* = 8.2 × 10^−9^). On the other hand, we found significant enrichment of cellulose synthase function in the set of genes that were down-regulated in response to parental wounding and overlapping with sRNA loci (*p* = 3.6 × 10^−5^).

To look for a possible direct association between wound-induced sRNA production and transgenerational differential gene expression, we compared the LFC of sRNA abundances at siRNA and nd-sRNA loci that were directly overlapping with or nearby differentially expressed genes. We found that sRNA loci near transgenerationally up-regulated genes had lower sRNA production in wounded leaves 72 h after wounding. Interestingly, this pattern is highly significant for sRNA loci nearby but not overlapping genes (<5 kb) (Table 2) but not for sRNA loci that directly overlap genes (Table 2) (Figure 5b). Within these nearby genes there is a significant sRNA locus type*direction change of expression effect (Table 2). nd-sRNA loci were more strongly associated with transgenerational gene expression responses. Patterns are similar but less extreme in wounded leaves 1 h after wounding (Table 2). The pattern was reversed at the 3rd leaf pair 72 h after wounding, with increased nd-sRNA abundances at loci with elevated gene expression in the next generation (Table 2).

In summary, we find that nd-sRNA abundances 72 h after leaf wounding is predictive of transgenerational methylation changes and expression shifts in nearby genes. In both cases the patterns reverse between the second and third leaves. In the second leaf pair, sRNA with increased abundances after wounding are associated with regions of lower methylation and expression in the progeny generation in response to wounding. In the third leaf pair, elevated sRNA expression is associated with transgenerational expression and methylation increases.

## 3. Discussion

Small, non-protein coding RNAs are diverse mobile signals that can regulate plant environmental responses [5,31]. In *M. guttatus*, altered trichome production in the progeny of wounded plants [54] has inspired a number of follow-up experiments designed to elucidate the underlying molecular mechanisms for the transmission of non-genetic information between generations. Past work has demonstrated the genotype specificity of this response [54,55], pointed at substantial differential gene expression associated with parental wounding [14,56] and suggested that the majority of differential methylation associated with parental wounding is associated with transposon CHH methylation but that genic CG methylation is also impacted [13]. Additionally, recent work has demonstrated transgenerational trichome plasticity transmitted through both the maternal and paternal germline, with maternal but not paternal transmission likely involving DNA methylation [9].

sRNAs may play a role in mediating these transgenerational responses [22]. sRNA work in *Mimulus* has been limited, although [68] described the major classes of *M. guttatus* miRNAs. Here, we sequenced the first *Mimulus* sRNAome in response to environmental variation. We test our *a prior* model for a role of sRNAs in transgenerational plasticity which includes (1) siRNA (or other sRNA) populations differ between leaf-wounded and control plants; (2) the proximity of sRNA loci increases the probability that protein coding regions will be differentially methylated or differentially expressed in progeny of leaf wounded plants; and (3) sRNA abundance, gene expression or DNA methylation differences in offspring of wounded compared to control plants correlate with sRNA profiles in the parent generation after leaf wounding. Results addressing each of these three points are discussed below. Generally, we find some evidence for a role of small RNAs in transgenerational plasticity, however our data contrast with the most parsimonious scenario involving the transmission of small RNAs between generations.

### 3.1. sRNA Responses at the Site of Leaf Wounding

sRNA responses were primarily limited to the wounded leaf, not persisting to later leaf stages or to offspring. Contrary to our prediction, only a handful of siRNA loci showed differential siRNA production in response to wounding. However, these few loci are clearly associated with wound response signaling. The most strongly up-regulated siRNA locus in wounded leaves produced sRNAs that map to *Jasmonic Acid Oxidase 1*. These loci produced approximately 4-fold higher siRNA levels in wounded compared to control tissue. A single siRNA molecule dominated at this locus (96% of reads). This siRNA was found in the same orientation as *JOX1*/Migut.F00689, mapping perfectly to the second intron. JOX1 inactivates Jasmonic Acid (JA) [69]. Therefore, the repression of *JOX1* through RNAi may function to increase JA levels and plant defense after leaf wounding. Methyl-Jasmonate elicits transgenerational resistance to herbivory in tomato and *Arabidopsis* [22]. Therefore, by altering JA concentration, this siRNA locus may mediate transgenerational phenotypic plasticity in *M. guttatus*.

We detected thousands of nd-sRNA loci with increased sRNA production in wounded leaves. sRNAs derived from tRNA fragments (tRFs) were especially abundant. Responsive tRFs varied with respect to their length, which anticodon and from which side of the mature tRNA they were derived. This variation suggests that tRF-mediated responses are likely regulated by a number of down-stream processes. tRFs appear to be primarily produced by RNases involving *both* dicer dependent and dicer independent mechanisms [70] function through diverse mechanisms to mediate post-transcriptional and translational silencing [47] and are ubiquitous across all forms of life [71]. Once thought of as simply responsible for mediating translation, it is becoming clear that tRNAs associated pathways play important roles in modifying life history traits. This includes both biotic and abiotic stress responses [72,73] and evidence for a tRF role in transgenerational plasticity across a diversity of taxa [51,52].

We found shifts in miRNA levels at ten loci in response to leaf wounding. Two miR166 loci had significantly reduced miRNA abundances in wounded leaves. Recent work demonstrates an important role for miR166 loci in abiotic stress response. miRNAs derived from miR166 loci mediate the abscisic acid (ABA) metabolic pathway [74,75]. Down-regulation of miR166 increases abscisic acid levels, potentially mediating local wound responses in these damaged leaves. Eight miRNA loci had significantly elevated miRNA levels in wounded leaves. Of these, the first and fourth most strongly up-regulated belong to the miR169 class. Increased expression from miR169 loci confers enhanced resistance to *Ralstonia solanacearum* bacteria in *Arabidopsis*. This resistance is associated with Clavata 1 and 2 proteins belonging to the LRR family [76]. In our analysis, *M. guttatus* miR169 targets were enriched for LRR domain proteins. Therefore, in response to leaf wounding, *M. guttatus* plants may increase expression of miR169 miRNAs, potentially reducing susceptibility to bacterial invasion at the site of leaf wounding.

### 3.2. sRNA Loci Are Proximal to Differentially Methylated and Expressed Genomic Regions

We found that sRNA loci are close to sequences (e.g., genes) that are differentially methylated or differentially expressed in response to parent leaf damage. As expected, given that siRNA can initiate transcriptional repression through RNA dependent DNA methylation [77], we found that genomic regions near 24 nt siRNA-producing loci were more likely to be differentially methylated in the progeny of wounded plants and that protein coding regions overlapping with sRNA-producing loci in general increased the probability that a gene would be differentially expressed in the progeny of wounded plants.

The majority of siRNA-associated genomic regions differentially methylated in response to parent wounding were transposable elements. Increased transposon activity after wounding may lead to RNA-induced DNA methylation through the production of 24 nt siRNAs [78]. Evidence for this model would include increased production of corresponding siRNAs in wounded or later tissues, which we do not observe. Therefore, it is possible that the siRNA response that mediates transgenerational methylation patterns is not induced in wounded leaf tissues but elsewhere in the plant. Additionally, we found that certain nd-sRNA loci were associated with increased DNA methylation in response to parental leaf wounding. Within this class of nd-sRNA loci, we found a positive association between LFC of sRNAs in the third leaf pairs and the change in methylation in the next generation. However, the mechanism responsible for this relationship is unclear.

Similar to patterns we observed linking DNA methylation in response to parental wounding and sRNA locus location, we found that protein coding regions overlapping with sRNA loci were more prone to transgenerational plasticity and that nd-sRNAs and siRNAs tended to overlap with transgenerationally up-regulated and down-regulated genes, respectively. The relationship between siRNA locus locations and down-regulated transcripts in the progeny of leaf wounded plants suggests that siRNAs produced from these loci may play a role in transgenerational epigenetic silencing. The association between up-regulated genes in the progeny of leaf wounded plants and nd-sRNA loci is less clear. Interestingly, Heat Shock proteins (HSPs) were significantly enriched in our set of genes that overlapped with nd-sRNA loci and exhibited increased expression in the offspring of wounded plants. Whether these nd-sRNA loci function specifically in mediating transgenerational activation of HSPs or this correlation is due simply to other factors, this class of genes and their associated sRNAs are candidates in future studies dissecting transgenerational plasticity in this system.

### 3.3. sRNA Responses Do Not Predict Transgenerational Plasticity

Our *a prior* model predicts that sRNA abundance, gene expression, and/or DNA methylation differences in the offspring of wounded compared to control plants correlate with sRNA profiles in the parent generation after leaf wounding. We found few sRNA loci with differential expression in the offspring of wounded plants and no evidence that sRNA responses are directly heritable between generations. In fact, our data suggest that shifts in sRNA abundance in the wounded leaves are negatively correlated with abundances in distant leaf pairs and in the next generation. We found weak associations between nd-sRNA (but not siRNA) abundances in wounded leaves and next generation gene expression and methylation responses. Even the weak association between parental nd-sRNA production and offspring gene expression and methylation were not clearly in the predicted direction; transgenerationally up-regulated and up-methylated genes tended to associate with decreased nd-sRNA abundance in the wounded parent leaves.

Among the four sRNA loci with differential sRNA abundances in response to parent leaf wounding, the only siRNA locus and the only nd-sRNA locus suggest that transgenerational plasticity at sRNA loci may involve calcium and retrograde signaling. The nd-sRNA locus overlaps a *Ycf2* homolog and the siRNA locus overlaps Calmodulin. *Ycf2* is involved in chloroplast protein uptake through interactions with HSPs [63]. Calmodulin is involved in calcium and retrograde signaling [61]. Chloroplasts are now understood to play a role in mediating stress responses through calcium signaling and these pathways provide a mechanism for transgenerational plasticity [79]. Therefore, our results suggest that calcium signaling and plastid interacting sRNAs may play a role in the *Mimulus* transgenerational response to wounding.

The sRNA population with the strongest correlated between parent and offspring abundances were tRFs. On the one hand, this population of nd-sRNAs exhibited a negative correlation between tRF production in wounded leaf tissue at the 2nd leaf pair in the parental generation and the at the 2nd leaf pair in the following generation. On the other hand, tRF production in 3rd, unwounded leaf pairs of the parental generation were positively correlated with tRF abundances in the next generation. However, no individual tRFs showed a pattern of statistically significant differential abundance in the progeny of leaf wounded plants. This result—shifts in whole classes of sRNAs between generations but no consistent patterns at the locus level—is in-line with results in apomictic Dandelion, where grandparental stress induces shifts in sRNA size classes but individual loci do not exhibit differential abundance [64]. Our tRF results are also in-line with recent data that tRF abundances are highly plastic in response to environmental cues and are affected by parental environment [51]. For example, Reference [53] showed that transgenerational tRF production in *Brassica rapa* can be significantly altered positively and negatively in response to heat. They additionally found that these tRFs were not likely to be found at different abundances in the parental generation in response to the same environmental signals [53,80]. Our results, coupled with previous transgenerational studies of tRFs in plant models support the conclusion that tRFs are highly plastic to environmental cues and can have abundances altered by experiences in past generations but that stable changes in their abundance are unlikely to mediate transgenerational epigenetic inheritance.

Our absence of evidence does not mean that inherited sRNAs play no role in transgenerational plasticity. It is possible and even likely that by combining data from independent experiments and with limited tissue sampling, we simply failed to detect some transgenerational associations between sRNA production, methylation and/or gene expression. Alternatively, the role that sRNAs have in mediating transgenerational plasticity is not dependent upon their continued presence in later developmental stages or in the next generation. There is substantial evidence for a multi-faceted response to wounding within a single generation that involves electrical signaling, reactive oxygen species, ion fluxes and hormones that act across a range of time periods and developmental stages [81]. These sRNA alternatives may be key players in transgenerational plasticity. However, if sRNAs are a critical mobile signal for transgenerational responses, recent discoveries support a complex interplay between various classes of small RNAs, DNA methylation and histone modifications [1,82]. These studies highlight the importance of proteins involved in each of these pathways in mediating transgenerational plasticity. For example, suppression of *MSH1* in *Arabidopsis* causes DNA methylation and histone deacetylation dependent epigenetic and phenotypic changes that require functioning sRNA directed methylation pathways in order to be transgenerationally heritable [83]. These authors propose a model for transgenerational memory that involves calcium signaling, hormone responses and histone modifications in addition to sRNA-mediated methylation repatterning to mediate transgenerational plasticity. Our work here corroborates this multifaceted epigenetic stress response involved in transgenerational plasticity and highlights a few specific pathways worthy of continued focus.

## 4. Materials and Methods

We randomly split selfed seed from a single *Mimulus guttatus* recombinant inbred line (RIL-94, F8 generation) into control and damage groups (as in Reference [14]). These lines are derived from a cross between a high elevation annual from Iron Mountain, Oregon (IM 767) and a coastal perennial from Point Reyes, CA, USA (PR) (Holeski, 2007). For the damage treatment, we punched two holes ca. 6 mm in diameter in both leaves of the second leaf pair at the developmental time point when the third leaf pair had expanded to 5 mm in length. While mechanical wounding does not elicit the full plant herbivory response, our previous work has demonstrated that this cue leads to substantial differential gene expression, methylation and trichome production in the following generation [9,13,14,54]. One hour and 72 h after wounding we collected second leaf pair tissue from 3 damaged and 3 control plants. Seventy-two hours after wounding, we collected third leaf pair leaves from damaged and control plants. Eight additional plants, 4 control and 4 damaged were grown until flowering, at which point plants were selfed and seed collected. Two months after seed harvesting, we grew up two individuals from each of these eight parent lines, all were grown under the same greenhouse conditions as our control and leaf-damaged plants in the prior generation. When plants reached the third leaf pair expanded stage we collected second leaf pair samples from each of these sixteen individuals.

### 4.1. RNA Extraction, Library Preparation and Read Mapping

We isolated total RNA from leaf tissue using Direct-zol RNA Mini-prep following the manufacturer’s instructions with in-column DNAse treatment. RNA from the 34 samples was sent to the Kansas University Medical Center Genome Core where it was analyzed using Agilent Bioanalyzer QC and sRNA libraries were prepared for each sample separately using the Illumina Small RNA library Prep kit. Reads were sequenced on one lane of an Illumina HiSeq2500 using high output mode. We used ShortStack 3.8.5 [84] to trim and map reads, as well as to identify sRNA clusters in the *M. guttatus* RIL94 reference genome [13]. Trimmed reads are available on the sequencing reads archive (https://www.ncbi.nlm.nih.gov/bioproject/PRJNA668761). After trimming and quality filtering reads, we were left with 230 million sRNA reads across all samples. Of these, 18.5% did not map to the *M. guttatus* reference genome, leaving 187 million mappable reads. Due to the small size of these reads, the majority of mappable reads (73%) mapped equally well to multiple regions of the *M. guttatus* genome. To account for this, we used the ShortStack “—mmap: u” setting to guide placement of multi-mapped reads using the local densities of uniquely mapped reads. This resulted in identification of 29,884 loci ranging in size from 15-11,725 bp (mean, 453 bp).

We further classified each locus into three categories (Figure 1b). Micro-RNA (miRNA) loci were identified using the fifteen-step successive approach criterion implemented in ShortStack. miRNAs derive from hairpin transcripts that, when processed, result in 20–24 nt sRNAs. miRNAs have been most extensively studied for their role in translational regulation involved in developmental transitions [85]. Dicer-derived sRNA loci were defined as those loci for which ShortStack-mapped read abundances were greater than 80% 20–24 nt length sRNAs. Due to their size class, these sRNAs are likely produced by dicer-dependent RNA interference (RNAi) machinery and are generally referred to as small interfering RNAs (siRNAs). Double stranded siRNAs function in both transcriptional and post-transcriptional regulation; their protein binding partners are largely dictated by their size (20–24 nt). Therefore, we established two distinct categories of dicer-derived sRNAs—loci producing 20–23 nt sRNAs and loci producing 24 nt sRNAs. Non-dicer derived sRNA (nd-sRNA) comprise a diverse mix of sRNA loci that were not identified as either miRNA or dicer-derived sRNA loci.

### 4.2. Differential Expression Calling

We used DeSeq2 [86] to explicitly test for differential production of sRNAs among tissues (Figure 1a,c) at each of the 14,936 sRNA loci with an RPM ≥2. We separated our differential expression analyses into miRNA loci, 20–23 siRNA loci, 24 nt siRNA loci and nd-sRNA loci due to substantial differences in the mean-variance relationships within each of these four groups. We ran DESeq2 “nbinomLRT” tests with treatment as the explanatory variable. For the offspring generation we included a nested maternal line effect in order to account for non-independence between siblings. We used the DESeq2 default Benjami-Hochberg to get an adjusted q-value. After we identified hundreds of differentially expressed nd-sRNAs, we used tRNADB-CE [87] to determine if these sRNAs mapped to any known tRNA genes. We identified the putative targets of miRNAs using psRNA [88] to identify the classes of genes likely regulated by these wound responsive miRNAs.

We tested for correlations of log fold change (LFC) in sRNA expression in response to wounding between the parent and offspring generations. Additionally, we specifically tested for correlation in LFC between parent and offspring generations for the subset of tRNA-derived sRNAs that were highly up-regulated in the parental generation in response to leaf wounding. For significance tests, we fit a general linear model considering LFC in the second leaf pair at 72 h after wounding, as well as in the third leaf pair at 72 h after wounding, as explanatory variables and LFC in the offspring as the response variable.

### 4.3. Downstream Statistical Analyses

After calculating LFCs of sRNA in response to wounding at each of our timepoints, as well as calling statistical significance, we tested associations in genomic locations between sRNA and previously identified gene methylation [13] and gene expression levels [14] in the offspring of wounded plants. First however, we looked for simple associations between sRNA locus overlap with protein coding regions and mean methylation or expression levels in control individuals taken from these experiments. We defined protein coding regions into three categories: genes not overlapping with an sRNA locus (>5 kb from an sRNA locus), genes overlapping with an siRNA locus (within 5 kb of an siRNA locus) and genes overlapping with an nd-sRNA locus (within 5 kb of an nd-sRNA locus). Using ANOVA, we tested for significant differences in mean gene expression among these three categories. Also using ANOVA, we tested for significant differences in CG, CHG and CHH gene methylation among these three categories; methylation being within exons or introns of protein coding regions.

To look for sRNA locus location associations with transgenerational methylation responses previously [13] detected we first asked if different classes of sRNA loci had different probabilities of overlapping with differentially methylated regions (DMRs). We used nominal logistic regression with the response variable whether a given sRNA locus overlapped with a previously identified DMR (coded as 1, overlapping or 0, non-overlapping), with explanatory factors of sRNA locus length, distance of sRNA locus to a protein coding region (O: Overlapping with a protein coding sequence; N: Near a protein coding sequence, within 5 kb; F: Far from a protein coding sequence, greater than 5 kb) and sRNA class (Dicer or Non-dicer).

We tested whether overlap of a protein coding region with a certain class of sRNA locus impacted the direction of methylation change in the progeny of leaf damaged plants. We fit a model that included all previously identified DMRs and tested whether their direction of change in methylation in the offspring of wounded plants corresponded with sRNA locus overlap. We used a Chi-square contingency table to identify associations between DMR up or down-methylation (coded as 1, up-methylated or 0, down-methylated in response to parent leaf damage) and overlap with sRNA loci (no sRNA locus overlap, siRNA locus overlap or nd-sRNA overlap).

Next, we tested for a relationship between the direction of change of sRNA production within the parent generation and the direction of methylation change in the offspring generation. We filtered our sRNA loci to those within 20 kb of a DMR and then ran separate general linear models by sRNA class (siRNA locus or nd-sRNA locus) with direction of change of methylation in the progeny generation as the explanatory variables and the LFC of sRNA levels as the response variable. We applied this general linear model to the 1 h and 72 h time points from the second leaf pair and the 72 h time point from the third leaf pair.

Lastly, using similar methods, we tested for associations between sRNA loci and gene expression changes in the progeny of wounded plants. We used a contingency table approach to test for association between sRNA locus overlap (siRNA loci, nd-sRNA loci or neither) with protein coding regions and differential gene expression following wounding in the parent generation as the response variable (gene upregulation in response to parental wounding coded as 1; gene downregulation in response to parental wounding coded as 0). Next, we fit a model with direction of gene expression change in the offspring generation as the response variable and mean log (expression) and overlap with sRNA locus (siRNA loci, nd-SRNA loci or neither) as explanatory factors. Finally, we looked for an association between the LFC of sRNAs and transgenerational changes in protein coding gene expression. We considered sRNA loci within 5 kb of protein coding regions and fit a model with sRNA LFC as the response variable and transgenerational mRNA change of expression (up, none or down), the class of sRNA locus (siRNA loci, nd-sRNA loci) and whether the sRNA directly overlapped or was within 5 kb but not overlapping the protein coding region.

## Figures and Tables

**Figure 1 ijms-21-07552-f001:**
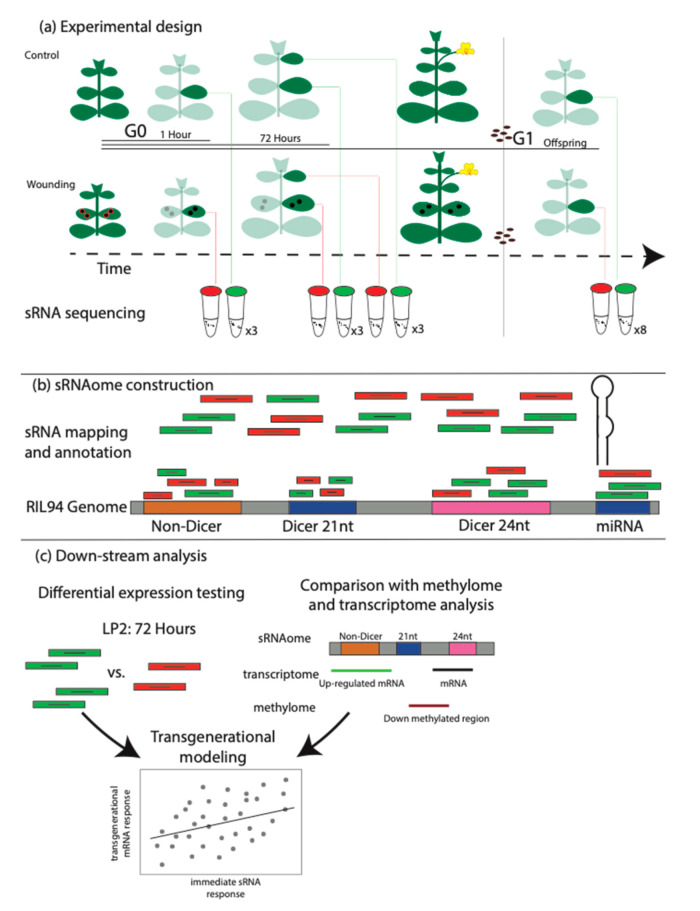
Experimental design and analytical methods. (**a**) Plants were germinated and placed into two experimental groups, one that received wounding on the second leaf pair and a control group. Leaf tissues from the wounded second leaf pair were collected 1 h after wounding (or control) and 72 h after wounding (or control). At the 72 h time point, leaf tissues from wounded and control plants were additionally collected from the third leaf pair which did not experience direct wounding. A subset of plants (wounded and control) were taken through to flowering, self-pollination and seed collection. This seed was germinated and progeny plants were propagated until the third leaf pair was expanded to 5 mm diameter at which time second leaf pair tissues were collected. (**b**) RNA was extracted, libraries prepared and sequenced from all tissue collections. Reads were mapped to a customized reference genome (RIL94, Colicchio et al., 2018 and annotated using ShortStack. (**c**) DEseq2 was used to determine differential abundances of sRNAs between developmental stages and wounding treatments. Additionally, the location of sRNA loci was compared with previously collected gene expression and DNA methylation data.

**Figure 2 ijms-21-07552-f002:**
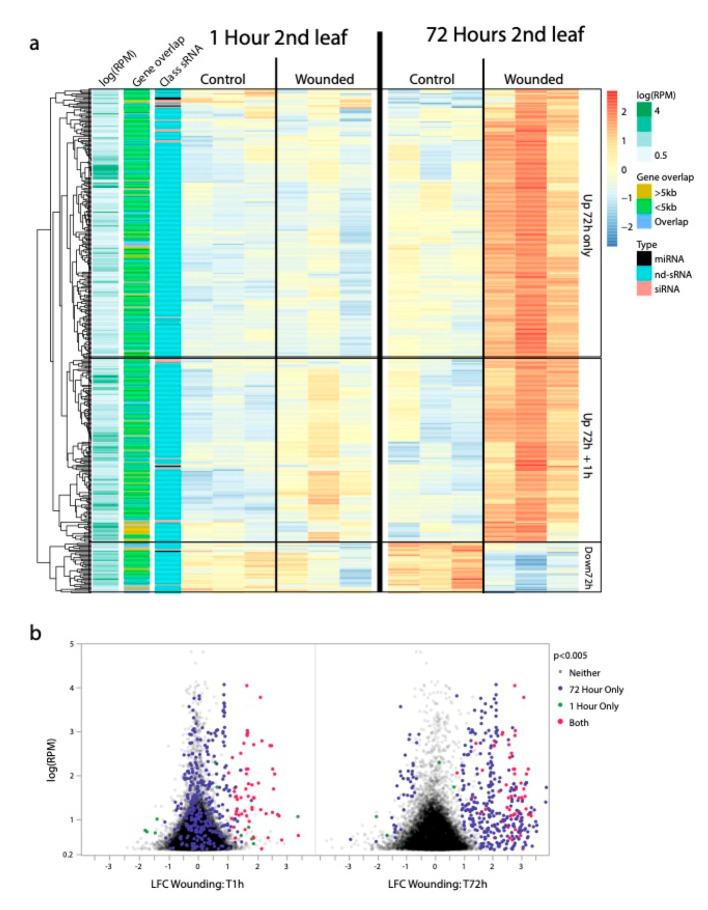
sRNA loci differentially expressed in response to leaf wounding (FDR < 0.05) (**a**) Clustered heat map showing patterns of differential sRNA production in wounded leaf tissues relative to control leaf tissues. The most striking differential expression is the increased abundance of non-dicer derived sRNAs (nd-sRNAs) that are < 5 kb from but not overlapping with protein coding regions. Many of these loci only exhibit elevated sRNA production at the site of leaf wounding (2nd leaf pair) 72 h after wounding, while a subset also show less extreme elevated sRNA production at the site of leaf wounding (2nd leaf pair) 1 h after wounding. (**b**) Volcano plot of log-fold changes in sRNA production from the 2nd leaf pair 1 h and 72 h after leaf wounding. A subset of sRNA loci show immediate increased sRNA production 1 h after leaf wounding. These loci tend to still have increased sRNA production 72 h after leaf wounding but are joined by more sRNA loci that did not exhibit differential sRNA production between wounded and control tissues 1 h after leaf wounding.

**Figure 3 ijms-21-07552-f003:**
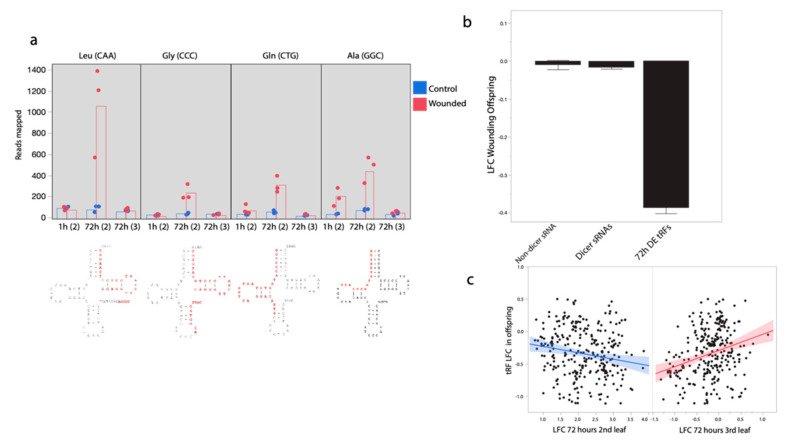
Differential production of tRNA-derived small RNAs (tRFs) in response to leaf wounding. (**a**) tRFs mapping to both the 5′ and 3′ end of tRNA loci but not the anticodons, show increases in tRF sRNA production in 2nd pair wounded leaves. Levels of tRF abundances from three tRNAs with the strongest pattern of differential tRF abundances are shown, as well as with significant increase in tRF abundances at both the 1 and 72 h timepoints. tRNA structure below with primary tRF sRNA fragment detected here in red. Leaf pair in parentheses Leu = Leucine, Gly = Glycine, Gln = Glutamine, Ala = Alanine. (**b**) tRF loci with increased expression 72 h after wounding in the wounded leaf, tend to have decreased expression in the following generation. (**c**) When modeled together, there is a negative relationship between log-fold changes (LFC) in the second leaf pair and LFC in the offspring for tRFs but a positive relationship between LFC in the third leaf pair and LFC in the offspring generation.

**Figure 4 ijms-21-07552-f004:**
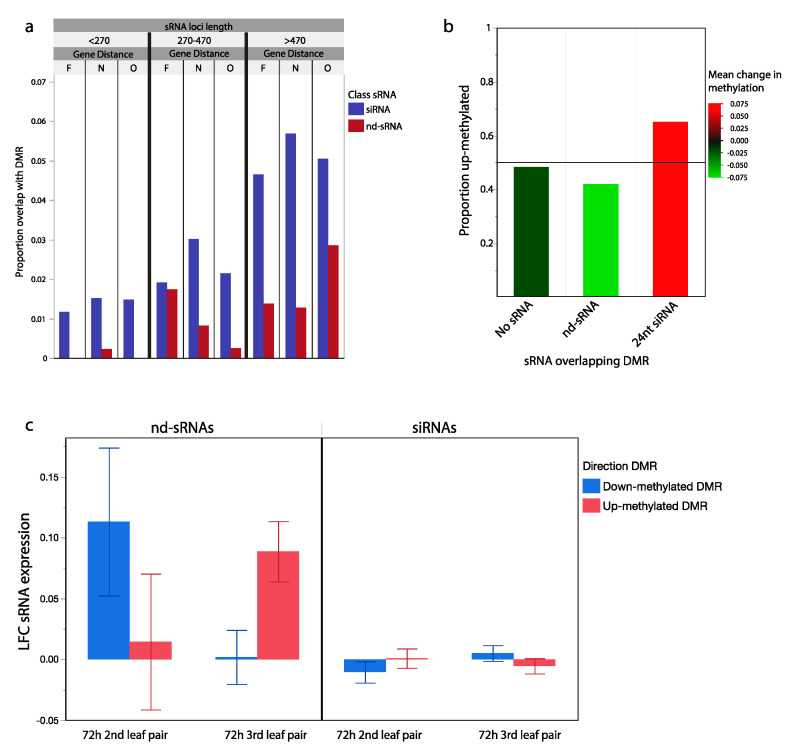
sRNA loci and transgenerationally differentially methylated genomic regions. (**a**) Proportion of siRNA and nd-sRNA loci that overlap with regions previously identified as differentially methylated in the offspring of wounded plants. sRNA locus types sorted into groups of similar size and similar distance from protein coding regions (Far, F: >5 kb, Near, N < 5 kb, Overlapping, O). Longer sRNA loci near but not overlapping with genes had the highest frequency of overlap with genomic regions differentially methylated in response to parent leaf wounding. siRNA loci overlap at higher frequency than nd-sRNA loci (*p* < 0.001) with regions differentially methylated in response to parent leaf wounding. (**b**) Differentially methylated regions overlapping with siRNA loci were more likely to have increased methylation in the progeny of wounded plants (0.66 up, 0.34 down), while those overlapping with nd-sRNA show the opposite pattern (0.42 up, 0.58 down) (*p* < 0.001, Chi Sq. = 75.21). Color-coded based on mean methylation change in each set of overlapping differentially methylated regions. (**c**) Differentially methylated regions overlapping with nd-sRNA loci show evidence of being associated with changes in sRNA abundances 72 h after leaf wounding. In the wounded 2nd leaf pair, there was an increase in sRNA abundance from sRNA loci overlapping with regions that were shown to be down-methylated in the next generation, while in the 3rd leaf pair there was an increase in sRNA abundance from sRNA loci in regions that were shown to be up-methylated in the next generation (*p* = 0.009). Error bars: +/− SE.

**Figure 5 ijms-21-07552-f005:**
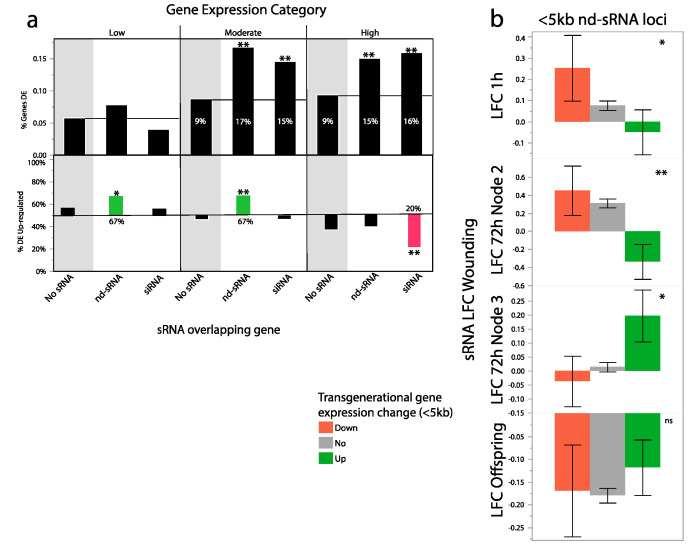
sRNA loci and transgenerationally differentially expressed genes. (**a**) Top: genes that overlap with sRNA loci are more likely to be differentially expressed than those that do not. Patterns of differential expression are similar for genes overlapping siRNA and nd-sRNA loci. Bottom: The direction of gene expression change varies by sRNA locus class. Lowly and moderately expressed genes that overlapped with nd-sRNA loci were more likely to be up-regulated in the progeny of wounded plants (up: 0.67, down: 0.33), while highly expressed genes that overlapped with siRNA loci were more likely to be down-regulated in the progeny of wounded plants (up: 0.20, down, 0.80). (**b**) Non-dicer sRNA plasticity to wounding tends to be negatively associated with transgenerational gene expression responses of genes near but not overlapping the sRNA loci (<5 kb). Genes that are down-regulated in the next generation tend to have increased sRNA abundance in the parents at 72 h in the wounded second leaf, while genes up-regulated in the next generation tend to have decreased sRNA abundance in the parents at the 72 h wounded second leaf. This same class of sRNAs shows similar but less extreme patterns in wounded leaf tissue at 1 h after wounding but an opposite pattern at 72 h in the unwounded 3rd leaf. ** *p* < 0.001, * *p* < 0.05.

**Table 1 ijms-21-07552-t001:** Summary statistics of expression and location of sRNA loci.

sRNA Class	Number of Identified Loci	Log(rpm)	Mean Locus Length	% Overlap with CDS ^4^	% Overlap with TEs ^5^
miRNA ^1^	44	1.4	227	9.3%	29.6%
siRNA ^2^	26,052	0.37	409	9.7%	89.9%
20 nt	10	0.46	200	0	66.7%
21 nt	226	0.78	601	45.3%	51.6%
22 nt	411	0.76	501	12.6%	78.7%
23 nt	50	0.39	254	12.5%	87.5%
24 nt	25,412	0.36	406	9.4%	90.4%
nd-sRNA ^3^	3775	0.66	774	62.7%	30.8%

^1^: Micro RNA, ^2^: Dicer-derived sRNA, ^3^: Non-dicer sRNA, ^4^: Coding Sequences, ^5^: Transposable Elements.

**Table 2 ijms-21-07552-t002:** Models linking sRNA LFC and mRNA transgenerational plasticity.

	1 h 2nd Leaf LFC			72 h 2nd Leaf LFC			72 h 3rd Leaf LFC		
Term	SS	F	*p*	SS	F	*p*	SS	F	*p*
mRNA Tgen Change ^1^ (2)	1.01	0.09	0.08	12.04	14.10	<0.0001	1.11	4.19	0.01
sRNA loci type ^2^ (1)	0.83	4.18	0.04	1.71	4.02	0.04	0.24	1.86	0.17
sRNA ^2^ x Tgen ^1^ (2)	1.20	3.01	0.05	10.43	12.21	<0.0001	0.76	2.87	0.06
Full Model: *n* = 7473	F = 4.4, *p* = 0.0005			F = 37.9, *p* < 0.00001			F = 1.9, *p* = 0.08		

^1^: Direction of mRNA expression change in the progeny of wounded plants (Up, Down, None) ^2^ sRNA locus type (siRNA, nd-sRNA).

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
