# Peer review of "Mimulus* sRNAs Are Wound Responsive and Associated with Transgenerationally Plastic Genes but Rarely Both"

_ijms, 2020, doi:10.3390/ijms21207552_

Round 1

Reviewer 1 Report

The authors examined possible roles of small RNAs in wound response and transgenerational reprogramming of gene expression. They found differentially expressed small RNAs – mostly tRNA fragments, and DNA methylation-dependent changes in gene expression in response to wounding, but surprisingly, not siRNA responses to wounding.

The study is remarkable and valuable by the choice of a less common model plant species and testing of clearly posed working hypotheses. The study follows the previous studies of authors on the same subject and is rich in novel data. I do not have any major points to be corrected.

Comments:

  1. Abstract should be shortened a bit without a substantial loss of the information.
  2. Results, lines 138-140 – symbols df, F, log10(rpm), and also some other abbreviations in the text should be explained upon their first use or rather in the list of abbreviations.
  3. The Discussion part is well elaborated. I only suggest to summarize the contents of its chapters and the main conclusions of the study to a single schematic picture, a kind of hypothetical model of regulatory factors and pathways involved in the wound response. This would substantially increase the impact of the study.

Author Response

Reviewer #1:

Comments and Suggestions for Authors

The authors examined possible roles of small RNAs in wound response and transgenerational reprogramming of gene expression. They found differentially expressed small RNAs – mostly tRNA fragments, and DNA methylation-dependent changes in gene expression in response to wounding, but surprisingly, not siRNA responses to wounding.

The study is remarkable and valuable by the choice of a less common model plant species and testing of clearly posed working hypotheses. The study follows the previous studies of authors on the same subject and is rich in novel data. I do not have any major points to be corrected.

We thank reviewer 1 for the kind comments.

Comments:

1. Abstract should be shortened a bit without a substantial loss of the information.

We have reduced the length of the abstract without any loss of information.

2. Results, lines 138-140 – symbols df, F, log10(rpm), and also some other abbreviations in the text should be explained upon their first use or rather in the list of abbreviations.

We have added an abbreviations section to the text.

3. The Discussion part is well elaborated. I only suggest to summarize the contents of its chapters and the main conclusions of the study to a single schematic picture, a kind of hypothetical model of regulatory factors and pathways involved in the wound response. This would substantially increase the impact of the study.

4. To improve clarity, we have reorganized the discussion.  In the revised version, we have organized the discussion around the three testable a prior hypotheses addressed in the manuscript. We feel this new streamlined version highlights our major take home points as well as key areas for follow up work.  While we do like the idea of including a schematic, we do not feel that the work here guides a model for the mechanisms of transgenerational plasticity and worry a schematic might be misleading.

Reviewer 2 Report

The presented manuscript from Colicchio et al. examines how plant wounding affects the smallRNA composition of wounded and neighboring leaves and whether these changes appear in the next generation of plants. The authors generated and analyzed a huge dataset using full sRNA library sequencing data. The manuscript deals with the description of these results and draw some conclusions and directs possible further research paths. The overall quality of the manuscript is generally good, although I have a few issues, see them below:

Major issues:

1.

At some points the precise analysis of the data lacks background knowledge of Mimulus genetics. These data and more tools (mutants, etc.) are available in Arabidopsis. Please explain the readers in the Introduction, why Mimulus guttatus is a better plant model for the performed research than Arabidopsis.

2.

I felt the Discussion chapter rather flatulent. A more compact text would focus on the main messages of the manuscript. I also found redundant parts which hide important points. For example, although the subject mentioned several times it is not clear what is the extent and importance of transgenerational transfer in the examined cases.

3.

Similarly to the Discussion, the work would benefit from shortening the abstract. Now it is too long. It should focus on the core findings.

4.

I really miss verification of the results. It is essential to check the expression change of genes/RNAs which were identified in a large scale (full genome, RNAome, etc.) approach. The targeted confirmation of the most interesting hits (e.g. miRNA_166, etc.) and the sRNA target gene expression would considerably strengthen the reliability of the results.

Minor issues:

Line 157: Typo in propagated.

Fig 4C: What do the error bars show?

Fig 4B and C: Are the differences significant? Can this be tested with statistical analysis?

Fig 5: What do the asterisks mean?

Table 2: Please provide explanations for the abbreviations.

Author Response

Review #2:

Comments and Suggestions for Authors

The presented manuscript from Colicchio et al. examines how plant wounding affects the smallRNA composition of wounded and neighboring leaves and whether these changes appear in the next generation of plants. The authors generated and analyzed a huge dataset using full sRNA library sequencing data. The manuscript deals with the description of these results and draw some conclusions and directs possible further research paths. The overall quality of the manuscript is generally good, although I have a few issues, see them below:

 We thank reviewer 2 for the kind words.

Major issues:

1.

At some points the precise analysis of the data lacks background knowledge of Mimulus genetics. These data and more tools (mutants, etc.) are available in Arabidopsis. Please explain the readers in the Introduction, why Mimulus guttatus is a better plant model for the performed research than Arabidopsis.

We have revised the second to last paragraph of the introduction highlighting Mimulus as a model system, and making a case for its study here.

2.

I felt the Discussion chapter rather flatulent. A more compact text would focus on the main messages of the manuscript. I also found redundant parts which hide important points. For example, although the subject mentioned several times it is not clear what is the extent and importance of transgenerational transfer in the examined cases.

We took this point to heart and reorganized the discussion section to improve clarity. In the revised version, we have organized the discussion around the three testable a prior hypotheses addressed in the manuscript. We feel this new streamlined version highlights our major take home points as well as key areas for follow up work. 

3.

Similarly to the Discussion, the work would benefit from shortening the abstract. Now it is too long. It should focus on the core findings.

We have reduced the length of the abstract without any loss of information.

4.

I really miss verification of the results. It is essential to check the expression change of genes/RNAs which were identified in a large scale (full genome, RNAome, etc.) approach. The targeted confirmation of the most interesting hits (e.g. miRNA_166, etc.) and the sRNA target gene expression would considerably strengthen the reliability of the results.

We appreciate this point, but qPCR verification is outside the scope of this study for the following reasons. qPCR validation of the gene expression results integrated into this study was full carried out in our previous paper (doi: 10.1111/nph.13081). Only sRNA validation of miRNA levels is possible and this, compared to nd-sRNA abundances, was a minor component of our results. Developing chemistry to undertake the miRNA validation will not, we feel, add to the strength of our conclusions.

Minor issues:

Line 157: Typo in propagated.

Corrected

Fig 4C: What do the error bars show?

Standard error, this is now included in figure legend.

Fig 4B and C: Are the differences significant? Can this be tested with statistical analysis?

Yes, these are included in the text, lines 371 and 407 and were now added to the figure legend.

Fig 5: What do the asterisks mean?

Significance. Added to figure legend.

Table 2: Please provide explanations for the abbreviations.

An explanation of the abbreviations has been added to figure legend.

Round 2

Reviewer 2 Report

The authors did a precise job and performed all required changes. I do believe these changes improved the clarity of the manuscript and resulted in better reading experience.